# Bounding boxes for weakly supervised segmentation: Global constraints get close to full supervision

**Hoel Kervadec**                                                    HOEL@KERVADEC.SCIENCE
*ÉTS Montréal*

**Jose Dolz**
*ÉTS Montréal*

**Shanshan Wang**
*Shenzhen Institutes of Advanced Technology*

**Eric Granger**
*ÉTS Montréal*

**Ismail Ben Ayed**
*ÉTS Montréal*

## Abstract

We propose a novel weakly supervised learning segmentation based on several global constraints derived from box annotations. Particularly, we leverage a classical tightness prior to a deep learning setting via imposing a set of constraints on the network outputs. Such a powerful topological prior prevents solutions from excessive shrinking by enforcing any horizontal or vertical line within the bounding box to contain, at least, one pixel of the foreground region. Furthermore, we integrate our deep tightness prior with a global background emptiness constraint, guiding training with information outside the bounding box. We demonstrate experimentally that such a global constraint is much more powerful than standard cross-entropy for the background class. Our optimization problem is challenging as it takes the form of a large set of inequality constraints on the outputs of deep networks. We solve it with sequence of unconstrained losses based on a recent powerful extension of the log-barrier method, which is well-known in the context of interior-point methods. This accommodates standard stochastic gradient descent (SGD) for training deep networks, while avoiding computationally expensive and unstable Lagrangian dual steps and projections. Extensive experiments over two different public data sets and applications (prostate and brain lesions) demonstrate that the synergy between our global tightness and emptiness priors yield very competitive performances, approaching full supervision and outperforming significantly DeepCut. Furthermore, our approach removes the need for computationally expensive proposal generation. Our code is shared anonymously.

**Keywords:** CNN, image segmentation, weak supervision, bounding boxes, global constraints, Lagrangian optimization, log-barriers

## 1. Introduction

Semantic segmentation is of paramount importance in the understanding and interpretation of medical images, as it plays a crucial role in the diagnostic, treatment and follow-up of many diseases. Even though the problem has been widely studied during the last decades, we have witnessed a tremendous progress in the recent years with the advent of deep con-

volutional neural networks (CNNs) (Litjens et al., 2017; Ronneberger et al., 2015; Rajchl et al., 2016; Dolz et al., 2018). Nevertheless, a main limitation of these models is the need of large annotated datasets, which hampers the performance and limits the scalability of deep CNNs in the medical domain, where pixel-wise annotations are prohibitively time-consuming. Weakly supervised learning has gained popularity to alleviate the need of large amounts of pixel-labeled images. Weak labels can come in the form of image tags (Pathak et al., 2015), scribbles (Lin et al., 2016), points (Bearman et al., 2016), bounding boxes (Dai et al., 2015; Khoreva et al., 2017; Hsu et al., 2019) or global constraints (Jia et al., 2017; Kervadec et al., 2019b). A common paradigm in the weakly supervised learning setting is to employ weak annotations to generate *pseudo-masks* or *proposals*. These proposals are ''fake'' labels, which are generated iteratively to refine the parameters of deep CNNs, thereby mimicking full supervision. Unfortunately, as discussed in several recent works (Tang et al., 2018; Kervadec et al., 2019b), proposals contain errors, which might be propagated during training, affecting severely segmentation performances. Furthermore, iterative proposal generation increases significantly the computation load for training. More recently, several studies investigated global loss functions, e.g., in the form of constraints on the target-region size (Pathak et al., 2015; Jia et al., 2017; Kervadec et al., 2019b; Bateson et al., 2019). This can be done by constraining the softmax outputs of deep networks, leveraging unlabeled data with a single loss function and removing the need for iterative proposal generation. Nevertheless, despite the good performances achieved by these works in certain practical scenarios, their applicability might be limited by the assumptions underlying such global constraints, e.g., precise knowledge of the target region size.

Among different weak supervision approaches, bounding box annotations are an appealing alternative due to their simplicity and low-annotation cost. In practice, bounding boxes can be defined with two corner coordinates, allowing fast placement and light storage. Furthermore, they provide localization-awareness, which spatially constrains the problem. This form of supervision has indeed become popular in computer vision to initialize shallow segmentation models, whose outputs are later used to train deep networks, as in full supervision (Dai et al., 2015; Papandreou et al., 2015; Khoreva et al., 2017; Pu et al., 2018). A naive use of bounding boxes amounts to generating pseudo-labels by simply considering each pixel within the bounding box as a positive sample for the respective class (Papandreou et al., 2015; Rajchl et al., 2016). However, in a realistic scenario, a bounding box also contains background pixels. To account for this, some advanced foreground extraction methods are employed. Particularly, the very popular GrabCut (Rother et al., 2004) is a standard choice to generate segmentation masks from bounding boxes, even though alternative approaches such as Multiscale Combinatorial Grouping (MCG) (Pont-Tuset et al., 2017) were recently used for the same purpose (Dai et al., 2015).

**Contributions:** We propose a novel weakly supervised learning paradigm based on several global constraints derived from box annotations. First, we leverage the classical tightness prior in (Lempitsky et al., 2009) to a deep learning setting, and re-formulate the problem by imposing a set of constraints on the network outputs. Such a powerful topological prior prevents solutions from excessive shrinking by enforcing any horizontal or vertical line within the bounding box to contain, at least, one pixel of the foreground region. Furthermore, we integrate our deep tightness prior with a global background emptiness constraint,

guiding training with information outside the bounding box. As we will see in our experiments, such a global constraint is much more powerful than standard cross-entropy for the background class. Our optimization problem is challenging as it takes the form of a large set of inequality constraints, which are difficult to handle in the context of deep networks. We solve it with sequence of unconstrained losses based on a recent powerful extension of the log-barrier method (Kervadec et al., 2019c), which is well-known in the context of interior-point methods. This accommodates standard stochastic gradient descent (SGD) for training deep networks, while avoiding computationally expensive and unstable Lagrangian dual steps and projections. Extensive experiments over two different public data sets and applications (prostate and brain lesions) demonstrate that the synergy between our global tightness and emptiness priors yield very competitive performances, approaching full supervision and outperforming significantly DeepCut (Rajchl et al., 2016). Furthermore, our approach removes the need for computationally expensive proposal generation.

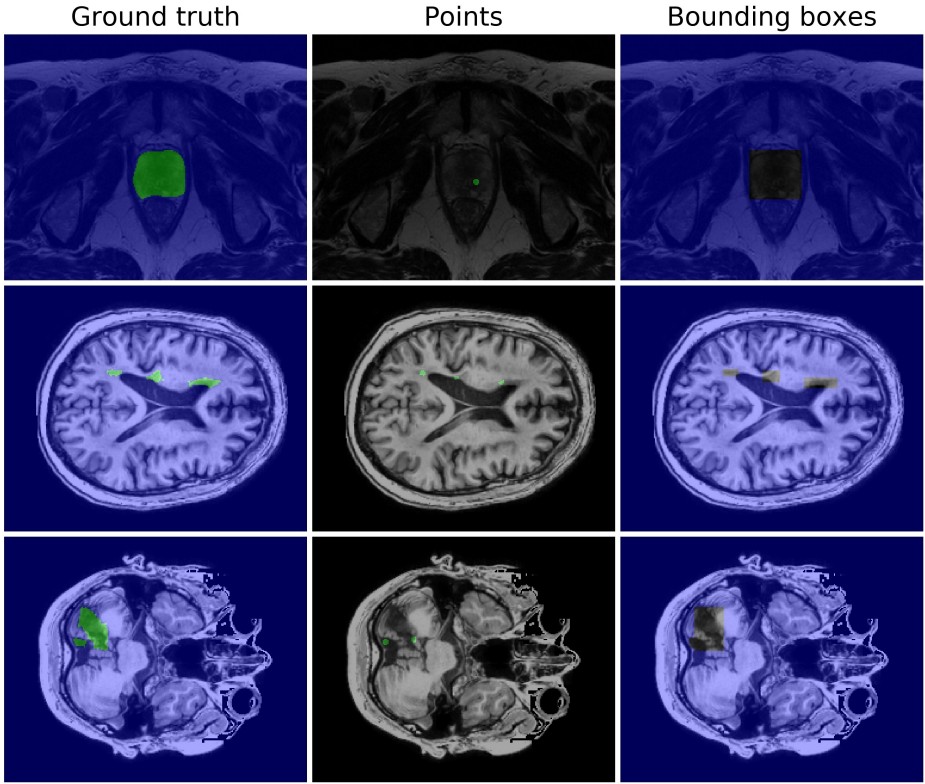

Figure 1: Example of weak labels on two different tasks: prostate segmentation and stroke lesion segmentation.

## 2. Related works

**Weakly supervised medical image segmentation.** Despite the increasing interest in weakly supervised segmentation models in the computer vision community, the literature

on these models in medical imaging remains scarce. The authors of (Qu et al., 2019) leverage point annotations in the context of histopathology images. From labeled points, they derived additional information in the form of a voronoi diagram, so as to generate coarse labels for nuclei segmentation. Their objective function integrated the cross-entropy with coarse labels and the conditional random field (CRF) loss in (Tang et al., 2018). Similarly to previous works in computer vision, (Nguyen et al., 2019) used classification activation maps (CAMs) derived from the networks as a pseudo-masks to train a CNN in a fully supervised manner. To constrain the location of the target, they employed an Active Shape Model (ASM) as a prior information. Nevertheless, this method presents two limitations. First, as in similar works, inaccuracies of the pseudo-masks may lead to sub-optimal performances. Second, the ASM is tailored to this specific application, as its generation for novel classes is dependent on the segmentation masks. More recently, (Wu et al., 2019) proposed to refine the generated CAM with attention, with the goal of generating more reliable pseudo-masks. Alternatively, other recent methods investigated how to constrain network predictions with global statistics, for instance, the size of the target region (Jia et al., 2017; Kervadec et al., 2019a,b; Bateson et al., 2019). This type of prior information can be imposed as equality (Jia et al., 2017) or inequality (Kervadec et al., 2019b; Bateson et al., 2019) constraint. Although such constrained-CNN predictions achieved outstanding performances in a few weakly-supervised learning scenarios, their applicability remains limited to certain assumptions.

**Bounding box supervision.** Most CNN-based methods under the umbrella of bounding-box supervision fall under the category of proposal-based methods. In these approaches, the bounding box annotations are exploited to obtain initial pseudo-masks, or proposals, typically with a shallow segmentation method, e.g., the very popular GrabCut method (Rother et al., 2004). Then, training typically follows an iterative scheme, which involves two steps, one updating the network parameters and the other adjusting the pseudo-labels (Dai et al., 2015; Papandreou et al., 2015; Khoreva et al., 2017). To further refine the pseudo-labels generated at each iteration, several works (Rajchl et al., 2016; Song et al., 2019) used the popular DenseCRF (Krähenbühl and Koltun, 2011) or other heuristics. While this might be very effective on some datasets, DenseCRF typically assumes that all the training images have consistent and strong contrast between the foreground and background regions. Finding the optimal DenseCRF parameters[1] is difficult when the contrast of the object edge varies significantly within the same dataset. Moreover, the ensuing training is not end-to-end, as it still relies on a DenseCRF post-processing, even at inference time. Another drawback of those bounding-box based learning approaches – which is also shared by other proposal-based methods in general – is that early mistakes will re-enforce themselves during training. For example, in DeepCut (Rajchl et al., 2016), while the pseudo-labels cannot grow beyond the bounding box, the inner foreground may gradually disappear. More recently, Hsu et al (Hsu et al., 2019) employed a Multiple Instance Learning (MIL) framework to impose a tightness prior in the context of instance segmentation of natural images. Focusing on instance segmentation, the method used bounding boxes generated by R-CNN. In such MIL framework, positive bags are composed of box lines while negative bags correspond to

---

1. Several hyper-parameters controls the edge sensitivity of popular DenseCRF (Krähenbühl and Koltun, 2011), mostly $\theta_\beta$ and $\theta_\gamma$, but also $\omega_1, \omega_2$ and $\theta_\alpha$ to some extent.

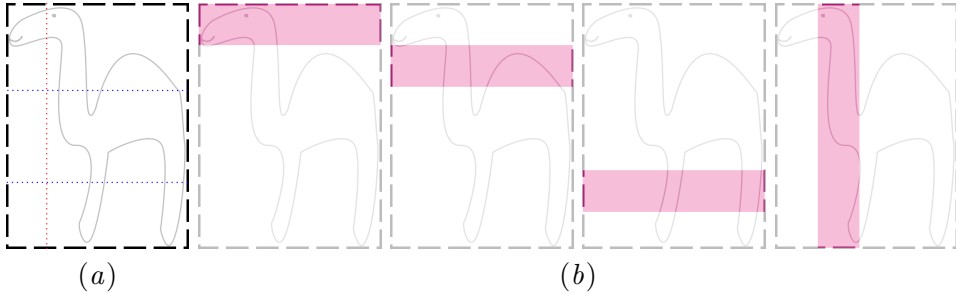

Figure 2: (a) Illustration of the tightness prior: any vertical (red) or horizontal (blue) line will cross at least one (1) pixel of the camel. (b) This can be generalized, where segments of width $w$ cross at least $w$ pixels of camel.

lines outside the box. The MIL loss function is defined so as to push the maximum predicted probability within each positive bag to 1, and the maximum predicted probability within each negative bag to 0. This MIL loss is integrated with a GridCRF loss (Marin et al., 2019) to ensure consistency between neighboring pixels. As many other works, the final predictions are refined with DenseCRF (Krähenbühl and Koltun, 2011).

## 3. Method

### 3.1. Preliminary notations

Let $X : \Omega \subset \mathbb{R}^{2,3} \to \mathbb{R}$ denotes a training image, and $\Omega$ its corresponding spatial domain. In a standard fully supervised setting, we can denote the training set as $\mathcal{D} = \{(X, Y)\}^D$, where $X \in \mathbb{R}^\Omega$ are input images and $Y \in \{0, 1\}^\Omega$ their corresponding pixel-wise labels. In the context of this work, however, labels $Y$ take the form of bounding boxes (as shown in Figure 1, third column). Thus, we use $\Omega_O$ and $\Omega_I$ to define the area outside and inside the bounding box, respectively, with $\Omega_O \cup \Omega_I = \Omega$. Let $s_{\boldsymbol{\theta}} \in [0, 1]^\Omega$ denote the probabilities predicted by the CNNs, where 0 and 1 represent background and foreground, respectively. In fully supervised setting, one would typically optimize the standard cross-entropy loss:

$$\min_{\boldsymbol{\theta}} \mathcal{L}_{\mathrm{CE}}(\boldsymbol{\theta}) := -\sum_{p \in \Omega} \left[ y_p \log(s_{\boldsymbol{\theta}}(p)) + (1 - y_p) \log(1 - s_{\boldsymbol{\theta}}(p)) \right].$$

### 3.2. Dealing with box annotations

**Certainty outside the box.** As shown in Figure 1, we certainly know that all pixels $p$ outside a given bounding box ($\Omega_O$) belong to the background. A straightforward solution would be to employ the cross-entropy, but only partially for each of those pixels outside the bounding box:

$$\mathcal{L}_{\mathrm{MCE}} := -\sum_{p \in \Omega_O} \log(1 - s_{\boldsymbol{\theta}}(p)).$$

Alternatively, notice that the size of the predicted foreground[2], when computed over the background pixels ($\Omega_O$), should be equal to zero. This gives the following global constraint for our optimization problem, which enforces that the background region is empty:

$$\sum_{p \in \Omega_O} s_{\boldsymbol{\theta}}(p) \leq 0. \tag{1}$$

We will refer to this constraint as the *emptiness constraint*, $\mathcal{L}_{EMP}$. $\mathcal{L}_O$ will denote either $\mathcal{L}_{\text{MCE}}$ or $\mathcal{L}_{\text{EMP}}$.

**Uncertainty inside the box.** While bounding box annotations provide cues about the spatial location of the target regions, pixel-wise information still remain uncertain. However, the bounding box can be further exploited to impose a powerful topological prior, referred to as *tightness prior* (Lempitsky et al., 2009). This global prior assumes that the target region should be sufficiently close to each of the sides of the bounding box. Therefore, we can expect that each horizontal or vertical line will cross at least one pixel of the target region (as illustrated in Figure 2), and for any region shape. Furthermore, we can regroup the lines into segments of width $w$, each containing $w$ lines. In this case, we can assume that at least $w$ pixels of the object will be crossed by the segment. Formally, we can write this as a set of inequality constraints:

$$\sum_{p \in s_l} y_p \geq w \qquad \forall s_l \in \mathcal{S}_L \tag{2}$$

where $\mathcal{S}_L := \{s_l\}$ is the set of segments parallel to the sides of the bounding boxes. This can be easily translated into inequality constraints on the outputs of the CNN, where the sum of the softmax probabilities for each segment should be greater or equal to its width. The set of segments $\mathcal{S}_L$ can be efficiently pre-computed; only the masked softmax sum is required during training.

### 3.3. Additional regularization: constraining the global size

The first two parts of the loss are biased toward opposed, trivial solutions: $\mathcal{L}_O$ trivial solution is to predict the whole image as background, while the easiest way to satisfy the tightness constraints (2) is to predict everything as foreground. But there is more information that we can exploit from the boxes: their total size gives an upper bound on the object size. We can also assume that a small fraction $\epsilon$ of the box belongs to the target region, which yield another lower bound. This takes the form of region-size constraint similar to (Kervadec et al., 2019b):

$$\min_{\boldsymbol{\theta}} \ \mathcal{L}_1(\boldsymbol{\theta}) + ... + \mathcal{L}_n(\boldsymbol{\theta}) \tag{3}$$

$$\text{s.t.} \ \epsilon|\Omega_I| \leq \sum_{p \in \Omega} s_{\boldsymbol{\theta}}(p) \leq |\Omega_I|.$$

---

2. Here we refer the size as the sum of the softmax probabilities, as it is easy to compute and differentiable. Therefore, it accommodates standard Stochastic Gradient Descent.

### 3.4. Lagrangian optimization with log-barrier extensions

Optimizing $\mathcal{L}_O$ with the constraints from sections 3.2 and 3.3 gives the following constrained optimization problem:

$$\min_{\boldsymbol{\theta}} \ \mathcal{L}_O(\boldsymbol{\theta}) \tag{4}$$

$$\text{s.t.} \ \sum_{p \in s_l} s_{\boldsymbol{\theta}}(p) \geq w \qquad\qquad \forall s_l \in \mathcal{S}_L$$

$$\text{s.t.} \ \epsilon|\Omega_I| \leq \sum_{p \in \Omega} s_{\boldsymbol{\theta}}(p) \leq |\Omega_I|.$$

This formulation involves a large number of competing constraints. Recent optimization works on constrained CNNs (Kervadec et al., 2019c) suggest that, in the case of multiple competing constraints, log-barrier extensions provide approximations of Lagrangian optimization in the form of sequences of unconstrained losses, which removes completely expensive and unstable primal-dual steps in the context of deep networks, handling the multiple constraints fully within SGD. Therefore, log-barriers can accommodate the interplay between multiple competing constraints, unlike naive penalty-based methods. These desirable properties are consistent with well-established interior-point and log-barrier methods in convex optimization (Boyd and Vandenberghe, 2004).

For an inequality constraint in the form of $z \leq 0$, the log-barrier extension can be defined as follows:

$$\tilde{\psi}_t(z) = \begin{cases} -\frac{1}{t}\log(-z) & \text{if } z \leq -\frac{1}{t^2} \\ tz - \frac{1}{t}\log(\frac{1}{t^2}) + \frac{1}{t} & \text{otherwise,} \end{cases} \tag{5}$$

where $t$ is a parameter that *raise* the barrier over time (i.e., during training). The main difference with a penalty (such as $\max(0, z)^2$, used by (Kervadec et al., 2019b)) is that (5) acts as a *barrier* even when the constraint is satisfied ($z \leq 0$), with a gradient getting more aggressive when approaching constraint-violation boundary. This makes the training more stable, and prevents already satisfied constraints from being violated during the next training epochs. Using a penalty could oscillate, alternating between zero and a high-penalty values (Kervadec et al., 2019c).

### 3.5. Final model

Using the log-barrier extension, we obtain the final unconstrained optimization problem, which can be optimized with standard SGD:

$$\min_{\boldsymbol{\theta}} \ \mathcal{L}_O(\boldsymbol{\theta}) + \lambda \left[ \sum_{s_l \in \mathcal{S}_L} \tilde{\psi}_t \left( w - \sum_{p \in s_l} s_{\boldsymbol{\theta}}(p) \right) \right]$$
$$+ \tilde{\psi}_t \left( \epsilon|\Omega_I| - \sum_{p \in \Omega} s_{\boldsymbol{\theta}}(p) \right) + \tilde{\psi}_t \left( \sum_{p \in \Omega} s_{\boldsymbol{\theta}}(p) - |\Omega_I| \right). \tag{6}$$

$\lambda$ is a real number balancing the tightness prior with respect to the other parts of the loss. Notice that all log-barrier extensions $\tilde{\psi}_t$ use the same $t$, with a common scheduling strategy for $t$. This limits the number of hyper-parameters and simplifies the model.

## 4. Experiments

### 4.1. Datasets and evaluation

We evaluate our method on two different tasks: prostate segmentation in MR-T2 and brain lesion segmentation in MR-T1. Among these tasks, lesion segmentation is particularly challenging, due to the heterogeneity of the lesions and high imbalance in the number of foreground and background pixels.

**Prostate segmentation on MR-T2.** The first dataset that we use was made available at the MICCAI 2012 prostate MR segmentation challenge[3] (Litjens et al., 2014). It contains the transversal T2-weighted MR images of 50 patients acquired at different centers, with multiple MRI vendors and different scanning protocols. The images include patients with benign diseases, as well as with prostate cancer. Images resolution ranges from $15 \times 256 \times 256$ to $54 \times 512 \times 512$ voxels, with a spacing ranging from $2 \times 0.27 \times 0.27$ to $4 \times 0.75 \times 0.75 \text{mm}^3$. We employed 40 patients for training and 10 for validation.

**Brain lesion segmentation on MR-T1.** We also evaluated the proposed method on the Anatomical Tracings of Lesions After Stroke (ATLAS) (Liew et al., 2018), an open-source dataset of stroke lesions. It contains 229 T1-weighted MR images, coming from different cohorts and different scanners. All the images have a resolution of $197 \times 233 \times 189$ pixels, with a spacing of $1 \times 1 \times 1$ mm. The annotations were done by a team of 11 experts, who received a standardized training. We retained 26 images for validation, while the rest were used for training.

**Evaluation.** To compare quantitatively the performances of the different methods, we employed the Dice similarity coefficient, a standard performance metric in medical image segmentation. In addition to the baseline models, we also perform comprehensive comparisons with DeepCut (Rajchl et al., 2016), whose learning setting is also based on bounding box annotations.

### 4.2. Implementation details

To evaluate our method under different settings, we experimented with a differnt network architecture for each task. We employ a residual version of the well-known UNet (Ronneberger et al., 2015) to segment the prostate, whereas ENet (Paszke et al., 2016) was a backbone architecture in the stroke lesion segmentation experiments. The models were trained with ADAM (Kingma and Ba, 2014), an initial learning rate of $5 \times 10^{-4}$ and a batch size of 4 for the prostate and 32 for stroke lesions. While we employed offline data augmentation (i.e., mirroring, flipping, rotation) to augment the PROMISE12 dataset, no augmentation was performed on the ATLAS dataset. The reason for this is the low number of images on the PROMISE12 dataset compared to ATLAS.

The log-barrier parameters were set following (Kervadec et al., 2019c), and were shared across all the log-barrier instances. We set $\lambda$ (from Eq. (6)) as 0.0001 for both datasets. The DenseCRF hyper-parameters are the same as in (Rajchl et al., 2016), and the proposals are updated every 10 epochs for PROMISE12, and every 5 epochs for ATLAS. We empirically

---

3. https://promise12.grand-challenge.org

found that changes on the width $w$ of the segments for the tightness constraints did not have a significant impact on the results. Therefore, $w$ was set to 5 in all the experiments.

All methods are implemented in PyTorch, with the exception of the DenseCRF (Krähenbühl and Koltun, 2011) which uses the Python wrapper PyDenseCRF [4]. To speed the proposal generation of DeepCut, the CRF inference is parallelized using the standard Python multiprocessing module, with a careful use of SharedArrays to avoid un-necessary and costly copies of arrays between the processes. The code is available online[5].

### 4.3. Sensitivity study on box-annotation precision

While the main experiments are performed on tight boxes (i.e., the gap between the target regions and the bounding-box sides is not significant), we perform additional experiments where a margin $m$ of 10 pixels was added on each side. This enables us to evaluate the robustness of each model to imprecise bounding-box placement. Robustness to placement is of significant importance, since perfect annotation of all bounding boxes might be unrealistic. Furthermore, robustness to imprecision also alleviates the problem of annotator subjectivity.

## 5. Results

### 5.1. Main experiment

The results of the segmentation experiments are reported in Table 1. We can observe that the proposed method consistently outperforms DeepCut (Rajchl et al., 2016) across the two datasets. The differences in performance range from 1% in the PROMISE12 dataset to 10% in the case of ATLAS. Furthermore, the results obtained from the two loss functions designed to deal with the background constraints indicate that the proposed global emptiness constraint is more effective in our setting. We hypothesize this is due to several factors. First, employing the emptiness constraint on background pixels results in all the constraint losses being on the same scale, which has very nice properties from an optimization perspective. Second, the imbalance nature of the segmentation task in the ATLAS dataset makes the use of the cross-entropy over all the background pixels a suboptimal alternative, forcing solutions that encourage empty segmentations. Finally, we can observe that the proposed method achieves performances comparable to full supervision, particularly in the task of stroke lesion segmentation. Using only a subset of the losses does not give optimal results, showing their synergy.

Figure 3 depicts the validation results over training of the different models. Even though DeepCut achieves similar results as the proposed approach in the PROMISE12 dataset, we can see that it is very unstable during training, as is the case generally for proposal-based methods. Additionally, its performance degrades over time. This effect is even more noticeable on the ATLAS dataset, where it collapses to empty segmentations after 25 epochs. This behaviour is a clear example of the instability of proposal-based methods, since we observed similar findings on the training images. More details about this issue are provided in Appendix A.

---

4. https://github.com/lucasb-eyer/pydensecrf

5. https://github.com/LIVIAETS/boxes_tightness_prior

| Method | PROMISE12 | ATLAS |
|---|---|---|
| | DSC | DSC |
| Deep cut (Rajchl et al., 2016) | 0.827 (0.085) | 0.375 (0.246) |
| Tightness prior | | |
|   w/ emptiness constraint | NA | 0.161 (0.145) |
| Tightness prior + box size | 0.620 (0.100) | 0.146 (0.134) |
|   w/ masked cross-entropy ($\mathcal{L}_{\mathrm{MCE}}$) | 0.774 (0.045) | 0.159 (0.203) |
|   w/ emptiness constraint ($\mathcal{L}_{\mathrm{EMP}}$) | **0.835 (0.032)** | **0.474 (0.245)** |
| Full supervision (Cross-entropy) | 0.901 (0.025) | 0.489 (0.294) |

Table 1: Results on the validation set for the proposed method, and the different baselines in both PROMISE12 and ATLAS datasets. The best results in the weakly supervised setting are highlighted in bold. NA means that the network didn't learn to segment anything meaningful.

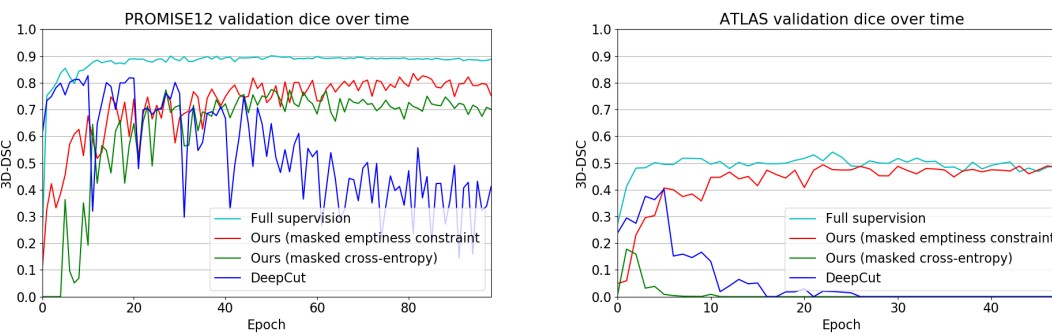

Figure 3: Evolution the validation DSC values over time for both PROMISE12 and ATLAS, and for different methods.

Qualitative segmentation results are depicted in Fig 4. We can observe how the proposed method with masked CE achieves satisfactory visual results on the prostate (first two rows), but fails to properly segment stroke lesions (last two rows). In contrast, when background segmentations are optimized with the proposed emptiness constraint, we observe how the segmentation results approach full supervision performance in both datasets. This is in line with the results reported in Table 1. On the other hand, DeepCut succeeds to segment the prostate but it is not able to obtain satisfactory segmentations for brain lesions. Looking closer at these segmentations, we can observe that they do not reliably follow the target boundaries. This can be explained by the fact that denseCRF assumes strong contrasts between foreground and background regions, which is not the case in many of these images. Furthermore, the results provided by denseCRF are sensitive to its hyper-parameters $\theta_{\beta}$, $\theta_{\gamma}$, $\omega_1$ and $\omega_2$, which control the edge sensitivity. Since the set of hyper-parameters were fixed across all the images in the whole dataset, it might happen that an optimal set of hyper-parameters for a given image performs sub-optimally for another image.

| Ground truth | Full supervision | Ours (emptiness constraint) | Ours (masked CE) | DeepCut |
|---|---|---|---|---|

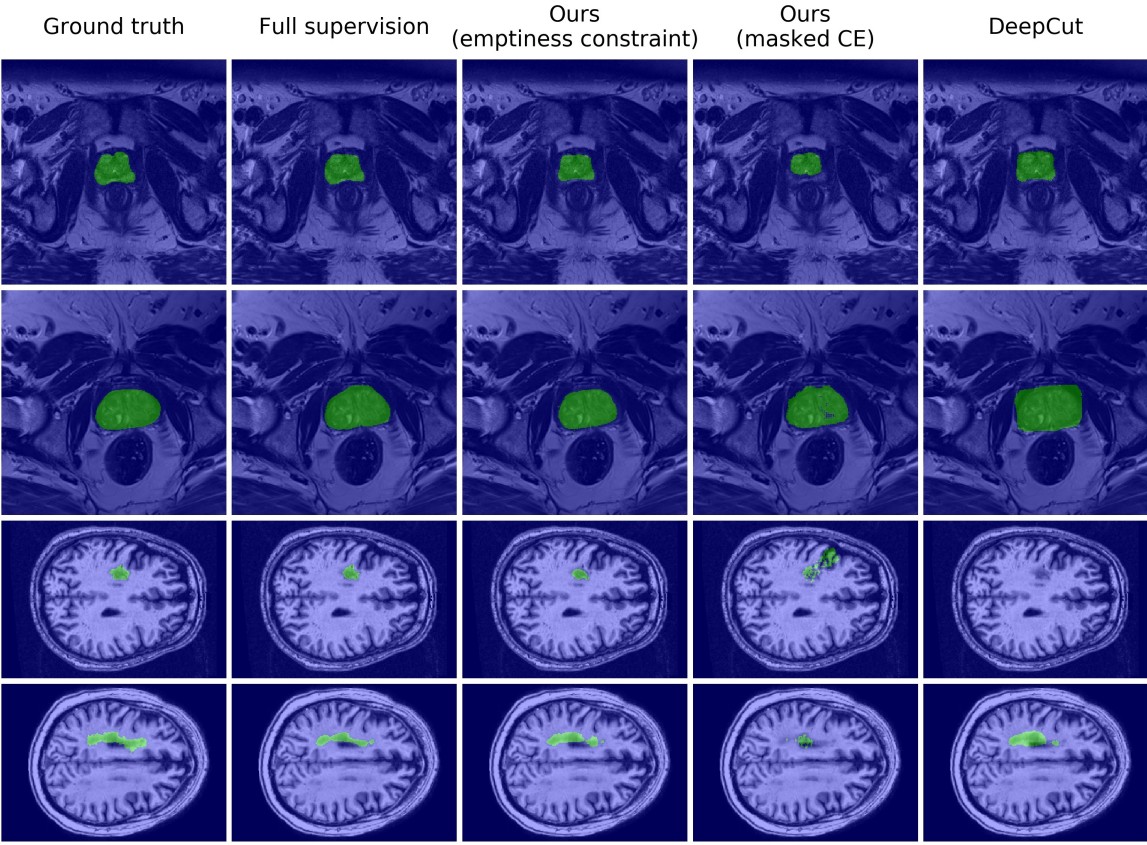

Figure 4: Predicted segmentation on the validation set for the two tasks.

## 5.2. Resilience to box imprecision

Results of the sensitivity study on the box precision are reported in Table 2. While all methods were able to reach similar performances when the bounding box annotation is nearly perfect (despite stability issues for some methods), their performance degrades as the margin between the region of interest and the borders of the bounding box increases. Specifically, if a margin $m$ of 10 pixels is added on each side, the performance of the proposed method only drops by 5%, in terms of DSC, whereas DeepCut performance decreases by 14%.

| Method | Margin=0 | Margin=10 |
|---|---|---|
| DeepCut | 0.827 (0.085) | 0.684 (0.069) |
| Ours (emptiness constraint) | **0.835 (0.032)** | **0.778 (0.047)** |

Table 2: Sensitivity study wrt. the box margins on the PROMISE12 dataset. Best results highlighted in bold.

Finally, the computational cost of the different methods is discussed in more details in Appendix B.

## 6. Conclusion

In this paper we proposed a novel weakly-supervised learning paradigm based on several global constraints, which are derived from bounding box annotations. First, the classical tightness prior is integrated into a a deep learning framework by reformulating the problem as a set of constraints on the outputs of the network. Second, a global background emptiness constraint is employed to enforce empty segmentations outside the bounding box, which is demonstrated to be more powerful than standard cross-entropy for handling the background class. Integration of such a large set of inequality constraints on deep networks represents a challenging optimization problem.

We solve it with sequence of unconstrained losses, which are based on a recent extension of the log-barrier method. Since this formulation accommodates standard stochastic gradient descent, it can be easily trained on deep networks. We performed comprehensive experiments on two public benchmarks for the challenging tasks of prostate and brain stroke lesion segmentation, and demonstrated that the proposed approach outperforms state-of-the-art approaches with bounding-box supervision. Furthermore, quantitative and qualitative results indicate that the proposed approach has the potential to close the gap between bounding-box annotations and full supervision in semantic-segmentation tasks.

The sensibility study showed that the proposed method is resilient to imprecision in the box tightness. Future works will investigate the use of 3D bounding boxes as annotations, which will make the corresponding 2D boxes looser. Such a workflow could further speed up the annotation process. The proposed framework could also be extended to 3D-CNN, by generating segments for the tightness prior along the three axes. Furthermore, our approach is also compatible with multi-class segmentation problems, even when bounding boxes of different classes overlap.

## Acknowledgments

This work is supported by the National Science and Engineering Research Council of Canada (NSERC), via its Discovery Grant program. We also thank NVIDIA for the GPU donation.

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

## Appendix A. DeepCut training instability

We investigated the generated pseudo-labels (as showed in Figure 5) by DeepCut, and the main culprit is when the proposal under-segment the object inside the box. This forces, at the next training step, the network to segment the object as background. This kind of conflicting feedback to the network (some other proposal label similar looking patches as foreground) makes the training unstable, and slowly skew the network toward empty predictions. This will cause the next batch of proposals to be even smaller, until the network outputs empty foreground for all the images.

| Ground truth | Initial box | 1st iteration | 2nd iteration | 4th iteration | 8th iteration |

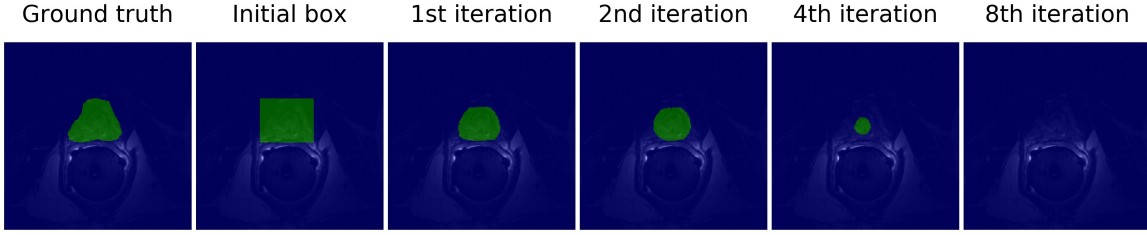

Figure 5: Progression of the pseudo-labels from DeepCut: only a few of those cases can make the training very unstable.

## Appendix B. Implementation and performances

Performances were measured on a machine equipped with an AMD Ryzen 1700X, 32GB of RAM (frequency did not affect speed) and an NVIDIA Titan RTX. They are reported in Table 3. The settings and hyper-parameters are the same as described in Section 4.2.

Most of the extra time introduced by our model comes from the naive log-barrier implementation that we used. Instead of leveraging `if/else` switch and code vectorization we used a standard Python `for` loop over all constraints. This could be improved using the recent PyTorch development of its JIT compiler. The width parameter of the segments will affect the overhead of our method: wider segments means less of them, which, in turns, results in less constraints to handle.

Notice that implementing the DenseCRF post-processing in a parallel and efficient fashion introduces a lot of software engineering uncommon in modern learning frameworks. While the DenseCRF implementation itself is highly efficient, it remains a single process that can handle only one image at a time. Performing it in parallel should be easy in theory, but is actually not very efficient with Python standard multiprocessing tools. In practice, all the arrays (containing either the image or probabilities) are pickled and copied across processes. Those back-and-forth copies can add up quickly and slow-down the processing substantially, on top of filling the computer memory more quickly. The solution is to carefully use SharedArray[6], which will contain all the batch in a single object. The sub-processed will read and write only a subset of those SharedArrays, corresponding to their assigned batch item.

---

6. Carefully, because they are not concurrency safe.

| Method | Time per epoch (s) | | Proposals update (s) | | Total (h) | |
|---|---|---|---|---|---|---|
| | Pr | At | Pr | At | Pr | At |
| Full supervision | 150 | 235 | - | - | 4.2 | 3.3 |
| Ours | 170 | 325 | - | - | 4.7 | 4.5 |
| DeepCut | 150 | 235 | 440 | 3120 | 6.6 | 11.9 |

Table 3: Comparison in training speed between the different methods on the two datasets, PROMISE12 (Pr) and ATLAS (At).

