# OpenReview forum: "Bounding boxes for weakly supervised segmentation: Global constraints get close to full supervision"
_MIDL.io/2020/Conference — MIDL 2020_

### Official Review · AnonReviewer2 · 2020-03-12
**new loss functions based on bounding box are introduced for weakly suppervised learning**

**Rating:** 3
**Confidence:** 4
**Recommendation:** Oral

**Summary:**

This paper introduces several new losses based on bounding box for weakly suppervised learning. Experiments on two medical image segmentation tasks show that the performance with the new loss approaches the suppervised version of the same model. Bounding-box a much cheaper labelling method than manually segmenting the whole target, especially for medical image, so it is valuable to study using bounding box to train segmentation models.  One weakness of this paper is that the baseline model is relatively weak.

**Strengths:**

The major strength of this paper is that it develops a novel way of utilizing bounding box as a weak supervision for medical image segmentation, and the two new losses introduced inside and outside the box looks reasonable. Experiments on two dataset show improvement over a previously published method DeepCut. The paper is well organized and written.

**Weaknesses:**

1.	The baseline method DeepCut is relatively old, especially when considering that deep learning is a fast-developing area. A series methods, including DeepCut and newer ones can be found at: https://github.com/JackieZhangdx/WeakSupervisedSegmentationList. I wonder if DeepCut is the state-of-the-art.

2.	Regarding the main experimental results. The DSC on PROMISE12 is fairly good but its improvement over DeepCut is small, and I wonder if this improvement is statistically significant. On the other hand, there is a big improvement of DSC over DeepCut on the ATLAS dataset, and the gap to the suppervised version is small, but the final DSC of 0.474 is too low to be meaningful in practice. Even the full supervision approach only achieved a DSC of 0.489. I wonder what’s the state-of-the-art result on this dataset.


**Detailed Comments:**

In constraints of formular (3), if we assume the bound box size is |Q|, and the target region size is epsilon*|Q| , why not constrain the target size in a range around  epsilon*|Q|, but in [epsilon*|Q|,|Q| ]?

**Justification Of Rating:**

Its valuable to study weak suppression by bounding box, especially in medical image segmentation tasks, because the manual pixel-level labeling is very expensive. This paper proposes new losses both inside and outside the bounding box, and reformulate the objective function to make it feasible for backpropagation optimization. Experiments shows that the losses are workable, but I refrain from giving strong accept because of the relatively weak baseline.

**Paper Type:**

methodological development

**Questions To Address In The Rebuttal:**

1. Why no more recent methods using bounding box are used as baseline.
2. What's the state-of-the-art results on the  ATLAS dataset.


**Special Issue:**

yes

---

> ### Author Response · Authors · 2020-03-28
> **Reply to reviewer 2**
>
> We thank the reviewer for the detailed comments of our paper and we really appreciate the positive feedback. As highlighted by the reviewer the paper is well organized and written. The main concerns are on the baseline employed and on the state-of-the-art performance on the ATLAS dataset. Please find below the detailed answers to these concerns.
>
>
>
> > Why no more recent methods using bounding box are used as baseline.
>
> We employed DeepCut as a baseline method as it has been the only method evaluated in the medical imaging domain, trained with bounding boxes. Even though there exist a few other works focusing on natural images, they present some challenges in the context of medical imaging. For example, many of these methods rely on external foreground/background detectors trained on fully labeled images. The work in [7] employs the same setting as DeepCut when training solely from bounding boxes. Furthermore, they enhance the initial image proposals obtained by GrabCut by resorting to the Multiscale Combinatorial Grouping (MCG) approach to generate additional segment proposals. This method, however, is trained on pixel-level annotations. In [8], authors use additional instance segmentation masks from the SBD dataset. [9] employs a subset of COCO dataset to train a FCN to detect class-agnostic object masks, which serve as image proposals of the categories in the other subset. The goal of these methods is to extract initial proposals to train segmentation networks in a fully supervised manner, which are iteratively refined. Since the natural colour images targeted by these methods are totally different from medical images, this may potentially result in substantial errors in the initial proposals, making them unusable.
> We did additional experiments and comparison, discussed in the top-level comment.
>
> [7] Simple Does it.
> [8] Pseudo-mask augmentated object detection.
> [9] Learning to Segment Every Thing.
>
>
>
> > What's the state-of-the-art results on the ATLAS dataset.
>
> Recent works on this dataset [3,4] have shown that fully supervised learning results in DSC values in the range between 0.48 and 0.53, which is similar to the results reported in this work.
>
> [3] Qi K et al. X-net: Brain stroke lesion segmentation based on depthwise separable convolution and long-range dependencies. MICCAI 2019.
> [4] Zhou, Y et al., D-unet: a dimension-fusion u shape network for chronic stroke lesion segmentation, IEEE/ACM transactions on computational biology and bioinformatics, 2019.
>
>
>
> > In constraints of formular (3), if we assume the bound box size is $|Q|$, and the target region size is $\epsilon|Q|$ , why not constrain the target size in a range around  $\epsilon|Q|$, but in $[\epsilon|Q|,|Q| ]$?
>
> Constraining around $\epsilon|Q|$ would require a rather precise $\epsilon$, valid across the whole dataset. With  $[\epsilon|Q|,|Q| ]$, we allow uncertainty in the target region size.

---

> ### Comment · AnonReviewer2 · 2020-03-29
> **adequate response to my concerns**
>
> With comparison to several more recent works and discussion of the state-of-the-art results on the second dataset, I’d like to upgrade the rating to Strong Accept.

---

### Official Review · AnonReviewer1 · 2020-03-13
**A novel loss function that leverages weakly annotated bounding box information to accurately perform semantic segmentation**

**Rating:** 4
**Confidence:** 4
**Recommendation:** Oral, Poster

**Summary:**

The authors propose a method to perform semantic segmentation of anatomical regions of interest from bounding box annotations (weakly supervised labeling). The approach utilizes two constraints: (i) a tightness prior and (ii) a background emptiness constraint. Segmentation results are presented for MRI prostate gland segmentation and MRI brain lesion segmentation. Results demonstrate that this approach (using inexpensive labeling) can achieve segmentation accuracies close to that of fully-supervised semantic segmentation (using expensive labeling).

**Strengths:**

-	Motivation is strong from a clinical perspective in that detailed annotations are expensive.
-	The background literature cited is comprehensive.
-	Testing on two different MRI datasets (for two different tasks: prostate gland segmentation and brain lesion segmentation) shows that the approach works well on different types of segmentation tasks.
-	Evaluation is compared to fully-supervised semantic segmentation and DeepCut methods.
-	Results show performance approaching that of fully-supervised semantic segmentation.
-	Preliminary results show that the method is relatively robust to errors in the bounding box segmentation.
-	The paper is well written and design choices are clearly explained.


**Weaknesses:**

-	Testing on n=10 subjects for prostate and n=26 for brain, is limited. Cross-validation studies would be more rigorous.
-	Contribution of the global size constraint (Sec. 3.3) is not quantified in the results.


**Detailed Comments:**

In my opinion, I think you can change the word “Ablation” study to “Sensitivity” study in this paper. Ablation studies involve knocking out certain methodological components to demonstrate the effect of each, but this is really a sensitivity analysis to bounding box error.

**Justification Of Rating:**

I think this is a strong paper that would be of interest to the MIDL community. The training loss function constraints are impactful, mostly novel contributions for medical image analysis. These initial results are encouraging and show segmentation performance approaching that of fully-supervised semantic segmentation while using weakly annotated (and therefore much less expensive labeling) data.

**Paper Type:**

methodological development

**Questions To Address In The Rebuttal:**

You might want to consider an ablation study of the global size constraint (from Sec. 3.3) to quantify the effect of this constraint relative to others.

**Special Issue:**

yes

---

> ### Author Response · Authors · 2020-03-28
> **Reply to reviewer 1**
>
> We thank the reviewer for the detailed comments of our paper and for having appreciated our novel contribution. The reviewer highlights the strong clinical motivation of our paper, and the clarity of the paper and reasonable motivation of design choices. The main concerns are related to limited number of testing images and the effect of the global size constraint. Please find below the detailed answers to these concerns.
>
>
>
> > Testing on n=10 subjects for prostate and n=26 for brain, is limited. Cross-validation studies would be more rigorous.
>
> We agree with this, and it was not performed due to time constraints. We will try to get those results by the time we need to submit the final manuscript. If not possible, we will do that in a future journal extension, which will not have those time constraints.
>
>
>
> > You might want to consider an ablation study of the global size constraint (from Sec. 3.3) to quantify the effect of this constraint relative to others.
>
> We did those experiments.
>
> The synergy of the three sub-losses is important. On the two tasks, the best results were reached (by a big margin) when we combine the three sub-losses. The values are (apologies for the formatting):
>
> ATLAS: tightness prior + global size: 0.146 DSC, tightness prior + emptiness constraints: 0.161 DSC.
>
> PROMISE12: tightness prior + global size: 0.620 DSC, tightness prior + emptiness constraints: NA (the network didn't succeed to learn. An aggressive search of the loss weight $\lambda$ might be able to reach decent results).
>
> The tightness prior + global size alone is capable to have partial results, but it tends to give high probabilities to pixels outside of the box. This explains the big improvement when we add the emptiness constraint.
>
>
>
> > In my opinion, I think you can change the word “Ablation” study to “Sensitivity” study in this paper. Ablation studies involve knocking out certain methodological components to demonstrate the effect of each, but this is really a sensitivity analysis to bounding box error.
>
> This is a very good suggestion. The manuscript will be updated accordingly.

---

### Official Review · AnonReviewer4 · 2020-03-15
**Possibly valuable idea, but the manuscript is hard to read and understand**

**Rating:** 2
**Confidence:** 4
**Recommendation:** Poster

**Summary:**

The authors propose to learn segmentation of medical images in 2D using a weakly supervised method. This method is based on bounding boxes annotation instead of pixel wise annotation, which are more expensive to obtain. They steer away from classical losses such as cross entropy loss, and methods such as deep cuts, and propose their own loss based on the intuition that bounding boxes are tight around the area of interest and the area outside the bounding box does not contain foreground pixels. Since their constraints are hard to optimise for, they resort to a log-barrier method which allows them to use their loss within a standard gradient descent optimisation technique.

**Strengths:**

The optimisation framework proposed by the paper, in order to deal with losses imposing inequality constraints on the outputs of the network, is interesting. It can be used in other applications beyond what's proposed here.

The results shown in the paper are good. It seems that the method is able to deliver a good improvement over other methods (Deep cut). A little more comparisons would have definitely helped, but the results already look relatively close to what one can achieve with full supervision.

**Weaknesses:**

The paper is pretty hard to understand due to its organisation and the way it is written. I personally had to read it multiple times in order to connect the dots and understand what had been done. It would have been much better to present the idea and the general intuition behind it at the very beginning, without introducing complex terms and details which could have been introduced later. The notation of the various formula and expression in the paper is not very standard and therefore confusing. I have checked most of the math, but I am not 100% sure everything is correct. The authors might want to double check everything and maybe align their notation to the notation used in other works.

It seems that the authors optimise for background emptiness, subject to some constraints pushing foreground within the bounding box. Foreground needs to have a minimum area and touch the bounding box boundaries in at least "w" points. Figure 2 is supposed to show the second constraint (foreground touches the bounding box in at least w points) but it actually does a terrible job explaining that. Please change figure 2.

One of the two proposed emptiness constraints (Eq. 1) prescribes the sum of the predictions outside the bounding box to be smaller-equal zero. The predictions are always positive though (so it can be zero, but not smaller than zero). This is true, unless the authors meant to indicate the "logits" (or network outputs before last activation) in the equation.

I am unsure the constraint "Uncertainty inside the box" is intuitively explainable. Why does the foreground need to touch the bounding box side in at least w points.

Constraining the global size relies on a manually supplied parameter epsilon which is decided by the user. It looks a bit arbitrary.

Finally, I personally disagree with the statement that pixel-wise segmentation is expensive. Pixel-wise segmentation is not expensive if it is mediated by interactive deep learning learning methods that do most of the work for the user. I believe that whoever still traces the ground truth segmentation without smart annotation tools by marking each individual pixel as background or foreground is just wasting time. Pixel-wise volumetric multi class segmentation can be achieved in seconds using smart annotation tools.

**Detailed Comments:**

The authors state that s_theta belonging to the interval [0, 1] over the image domain denotes the probabilities predicted by the CNN, where 0 and 1 represent background and foreground. I think that's the case for the labels, but the s_theta are subject to some form of thresholding in order to be converted from the continuous interval [0, 1] to discrete labels {0, 1}. Revise.

Figure 2 does not do a great job explaining the concept of "Uncertainty inside the box" constraint.

I have understood that the "Uncertainty inside the box" constraint prescribes that the bounding box perimeter is split in w segments and at least one pixel of each segment needs to be foreground. That's not what Eq. 2 enforces though. Eq. 2 states that at least w pixels across the whole bounding box contour need to be foreground and not that at least ONE pixel in each of the w bounding box contour segments needs to be foreground.

**Justification Of Rating:**

I believe the paper has merit, but I found the constraints proposed here heavily based on heuristics and user supplied parameters.

The presentation needs to be improved due to being currently unclear.

The proposed optimisation framework based on log-barrier method to make the loss optimisable using standard gradient descent technique is a great idea and can be used in multiple problems going beyond those presented in the paper.

I regard this paper as borderline. The results show improvement over other approaches and good results relative to fully supervised methods.



**Paper Type:**

methodological development

**Questions To Address In The Rebuttal:**

I would like the authors to revise the formulas and the figures. Adding clarity to the paper would just improve this work.

**Special Issue:**

no

---

> ### Author Response · Authors · 2020-03-28
> **Reply to reviewer 4 (part 2)**
>
> > Finally, I personally disagree with the statement that pixel-wise segmentation is expensive. Pixel-wise segmentation is not expensive if it is mediated by interactive deep learning learning methods that do most of the work for the user. I believe that whoever still traces the ground truth segmentation without smart annotation tools by marking each individual pixel as background or foreground is just wasting time. Pixel-wise volumetric multi class segmentation can be achieved in seconds using smart annotation tools.
>
> We thank the reviewer for this suggestion. We believe that weak supervision with bounding boxes is not in competition with interactive tools. In fact, it can provides inexpensive inputs for such tools. The wide interest in weakly supervised segmentation is evidenced by a good number of recent works in computer vision and medical imaging. In our opinion, the need is even more pressing in medical imaging, as full annotations require scarce clinical expertise and time. Also, we kindly disagree with the assumption that deep interactive segmentation may reduce the burden of obtaining annotations, at least for novel unseen scenarios. Recent works on deep interactive segmentation of medical images (e.g. [1-2]) show an important improvement with respect standard full-supervision settings. However, these methods still rely on pairs of images and the corresponding ground-truth labels during training, in addition to manual seeds placed by the user. Thus, in order to make these approaches work, costly manual annotations are still needed. Finally, even though we believe that deep interactive segmentation is an interesting field of research, the focus of the current paper is totally different, with a different scenario that does not contain a single fully labeled image.
>
> [1] Wang G et al. Interactive medical image segmentation using deep learning with image-specific fine tuning. TMI 2018.
> [2] Wang G et al. DeepIGeoS: a deep interactive geodesic framework for medical image segmentation. TPAMI 2018.
>
>
>
> > The authors state that $s_{\theta}$ belonging to the interval [0, 1] over the image domain denotes the probabilities predicted by the CNN, where 0 and 1 represent background and foreground. I think that's the case for the labels, but the $s_{\theta}$ are subject to some form of thresholding in order to be converted from the continuous interval [0, 1] to discrete labels $\{$0, 1$\}$. Revise.
>
> We use the probabilities when enforcing the constraints, as thresholding is non-differentiable. We could also add a temperature parameter to get closer to the vertices of the simplex, but this wasn't required in our application.

---

> ### Author Response · Authors · 2020-03-28
> **Reply to reviewer 4 (part 1)**
>
> We thank the reviewer for the detailed comments of our paper. The main concerns are on clarifying some technical details and on the structure and clarity of the paper. Please find below the detailed answers to these concerns.
>
>
>
> > The paper is pretty hard to understand due to its organisation and the way it is written. [...]
>
> We thank the reviewer for raising this concern. Nevertheless, we would appreciate if the reviewer could give more specific details that can lead to a better clarity, particularly since the other three reviewers emphasized in their reviews that the paper is well written, properly organized and easy to read, and that the method is formal and technically sound.
>
>
>
> > It seems that the authors optimise for background emptiness, subject to some constraints pushing foreground within the bounding box. Foreground needs to have a minimum area and touch the bounding box boundaries in at least "w" points. Figure 2 is supposed to show the second constraint (foreground touches the bounding box in at least w points) but it actually does a terrible job explaining that. Please change figure 2.
> > I am unsure the constraint "Uncertainty inside the box" is intuitively explainable. Why does the foreground need to touch the bounding box side in at least w points.
> > I would like the authors to revise the formulas and the figures. Adding clarity to the paper would just improve this work.
>
> Here following further clarifications of the tightness prior: The object does not need to touch the box edges at $w$ points. Rather, given that the box edges are close to the object, it becomes unlikely to find a vertical or horizontal line (wrt the box edges) that do not cross the bounded object at some point. This is true even for very convoluted shapes. This is what Figure 2 shows: you could "move" the blue or the red line, without finding a position where it doesn't cross the object.
>
> Once we notice that, we can generalize: if a single line cross the object at least once, then two lines must cross it at least twice, and so on and so forth. This gives the segments of width $w$, which are guaranteed to cross the object at least $w$ times.
>
>
>
> > One of the two proposed emptiness constraints (Eq. 1) prescribes the sum of the predictions outside the bounding box to be smaller-equal zero. The predictions are always positive though (so it can be zero, but not smaller than zero). This is true, unless the authors meant to indicate the "logits" (or network outputs before last activation) in the equation.
>
> Yes, the sum of probabilities is $\geq 0$. Constraining it to be also $\leq 0$ creates in effect a constraint to be equal to 0, which is what we want. We use an equivalent inequality formulation since this accommodate a powerful log-barrier framework (equality constraints are equivalent to 2 inequality constraints).
>
>
>
> > Constraining the global size relies on a manually supplied parameter epsilon which is decided by the user. It looks a bit arbitrary.
>
> The value of $\epsilon$ is chosen in a conservative manner. It can be viewed a weak supervision, as we know that at least a small fraction of the box belongs to the foreground. But it would be possible to relax $\epsilon$ to 0 as, in fact, the method doesn't rely heavily on it. Some other works, such as [1,2], used similar assumptions, and relied more heavily on it for their method to work.
>
> [1] Pathak, Deepak, Philipp Krahenbuhl, and Trevor Darrell. "Constrained convolutional neural networks for weakly supervised segmentation." ICCV 2015.
> [2] Khoreva, Anna, et al. "Simple does it: Weakly supervised instance and semantic segmentation." CVPR 2017.

---

### Official Review · AnonReviewer3 · 2020-03-19
**Well writen incremental paper on weakly supervised segmentation**

**Rating:** 4
**Confidence:** 4
**Recommendation:** Oral

**Summary:**

The paper proposes the use of a novel emptiness-constraint together with bounding boxes in weakly supervised CNN for image segmentation. It correctly builds upon previous works and attain a significant improvement over previous methods. The experimental analysis has a limited range, but shows the most important advantages of the proposed approach over previous ones.


**Strengths:**

It is very well written and structured. Concise and thorough. Very convincing reasoning and argumentation.

The description of the method is formal and technically sound. The code availability and the good description of the implementation seem to be complete enough for replication of the results by other researchers.

The results show improvement over previous methods.


**Weaknesses:**

The difference of the proposed approach to previous works by Kervadec and colleagues seem to be very small. At the same time, the experiments compare with Deep cut only, leaving unclear how the author’s results improve over the most similar previous approaches.

**Detailed Comments:**

The paper is easy to read and overall very clear. I just have some general comments below in addition to the weak points already mentioned above..

I feel authors are too much focused on improving in some metrics. While this is important, a more general view of the problem and the whys of the obtained results would enrich the paper.

Why using 2D bounding boxes, for example? Why not generalizing to the 3D domain? Who provides the bounding boxes in the workflow? A user provides them manually slice by slice? How did computation time compare with fully supervised techniques? How segmentation quality compares between fully supervised with fewer samples and weakly supervised with more samples? Etc..



**Justification Of Rating:**

The paper brings novelty and shows improved results over previous methods. It is well written, yet concise.
Minor issues, such as clarification on the improvement over Kervadec's previous works and a discussion on the impact of the proposed method on image analysis workflow can still be added in rebuttal phase.

**Paper Type:**

methodological development

**Questions To Address In The Rebuttal:**

The contribution over Kervadec et al.’s recent works is unclear. It sounds slightly incremental. The differences should be clarified so that one can assess the size of the contribution.

I suggest revising the conclusion. As it is now, it seems rather an abstract summary of results. A discussion about the results, why they are as they are, the limitations of the methods, the open possibilities to future work and the impact of the contribution on the application domain must be addressed.


**Special Issue:**

no

---

> ### Author Response · Authors · 2020-03-28
> **Reply to reviewer 3 (part 1)**
>
> We thank the reviewer for the detailed comments of our paper and the positive feedback. The reviewer highlights the very convincing reasoning and argumentation of our work, as well as the structure and clarity of the paper. The main concerns are on the main differences between the work in Kervaded et al [MedIA 2019] and the proposed work, and on clarifying some general aspects of our work. Please find below the detailed answers to these concerns.
>
>
>
> > The contribution over Kervadec et al.’s recent works is unclear
>
> While both works focus on constraining the output of CNNs, there exist some significant differences between them.
> - First, from a supervision perspective, [Kervadec et al., MedIA 2019] employed scribble annotations and a single region-size constraint at the image level. In contrast, this work leverage bounding box annotations, which much lager set of inequality constraints, including a topological tightness prior tailored to bounding box annotations. In the context of image segmentation (i.e., pixel-level classification), scribbles can be seen as semi-supervised annotations, where a subset of samples are annotated and labels are certain. Bounding boxes, however, fall within the category of weakly supervised annotations, as the information given is uncertain -- the bounding box includes both negative and positive samples.
> - Second, from an optimization perspective, [Kervadec et al. MedIA 2019] employed a penalty-based method as they had to deal with a single size  constraint. However, in optimization, it is well-known that penalties might have difficulties in dealing with a large set of competing constraints, as is the case of the topological tightness prior in our work. In fact, we experimented with penalty optimization in the case of our tightness prior, and observed that such a penalty approach leads to instabilities. Here, we resort to a powerful extension of the log-barrier approach, which is well-known in the context of interior-point methods. When dealing with a large number of constraints, log-barriers outperform significantly penalty-based methods, in terms of both accuracy and training stability.
> - Third, the constraints employed in this work are different and more realistic. [Kervadec et al., MedIA 2019] derived the size constraint from the annotated data. This scenarios achieves excellent performances, but assumes knowledge about the size of the target region, which might be unrealistic in several practical scenarios (as we do not have access to fully labeled data). In our work, the use of bounding boxes enable to leverage constraints in a more realistic setting. Our constraints go beyond precise region size and leverage topological and exclusion information derived specifically from box annotations.
>
>
>
> > I suggest revising the conclusion
>
> Upon rereading our conclusion, we agree with the reviewer, thank you for pointing that flaw. We will include more discussion, and also point to future possible works in the final version of the paper. Here is already a quick list of future extension and works that could be evaluated:
> - A study with 3D bounding cubes annotations (could be with a 2D or 3D segmentation network)
> - Studying a mix of fully and weakly annotated patients
> - Multi-class segmentation, with potentially overlapping boxes

---

> ### Author Response · Authors · 2020-03-28
> **Reply to reviewer 3 (part 2)**
>
> > Why using 2D bounding boxes, for example? Why not generalizing to the 3D domain?
>
> This is intended to be future work. A 3D bounding box would indeed make more sense and be faster to annotate with the right tools. The early results with less precise bounding boxes (Table 2) are actually encouraging, and suggest that a 3D box (which means less tight 2D boxes) could work.
>
>
>
> > Who provides the bounding boxes in the workflow? A user provides them manually slice by slice?
>
> Ideally, an annotator would provide bounding boxes of the object of interest, instead of task-consuming pixel-level labels. Since bounding boxes are not available for the analyzed datasets, we derive them from the ground truth annotations, with different levels of box margins (to mimic uncertainty in placing the bounding boxes).
>
>
>
> > How did computation time compare with fully supervised techniques?
>
> This is addressed in appendix B. The current slowdown is mostly due to the current log-barrier implementation, which is not vectorized (due to the if/else part of $\tilde{\psi}$, and forces to use a (slow Python) loop. We did a quick new comparison, implementing it in a vectorized fashion. On PROMISE12, full supervision is at 9.8 iter/s, ext log-barrier with a loop at 8it/s, and the new vectorized ext log-barrier at 9 iter/s. Some other efficiency gains could be achieved by tweaking the data structure used to store the bounding boxes and segments.
>
>
>
> > How segmentation quality compares between fully supervised with fewer samples and weakly supervised with more samples? Etc..
>
> We believe that this is a very interesting experiment. Particularly, comparing not only the performance between both settings, but also evaluating the time required to obtained the required annotations. We thank the reviewer for this suggestion and we keep that for future works.

---

### Author Response · Authors · 2020-03-28
**Top-level reply**

We thank the reviewers for the time they took to review thoroughly our paper. We are very pleased with the kind and constructive comments we received.
We will split our response in several comments. In this top-level comments, we will discuss some additional experiments we did within the week (mostly in reaction to Reviewer 2 concerns).

We studied two recent CVPR papers, and their feasibility to our application and supervision settings:
[1] Hu, Ronghang, et al. "Learning to segment every thing." CVPR. 2018.
[2] Song, Chunfeng, et al. "Box-driven class-wise region masking and filling rate guided loss for weakly supervised semantic segmentation." CVPR 2019.

[1] is an extension of Mask-RCNN ; which is already a chaining of R-CNN (CNN made for object detection, i.e. bounding boxes) and a small segmentation network. While the authors propose a method to learn segmentation with bounding box annotations (by training a transfer network between the bounding box parameters and the segmentation parameters), they still require a small subset of the dataset to be fully annotated. Therefore, it is not comparable to our method in the current setting, as we use only bounding boxes. However, it will be a very interesting paper to compare to for a future extension, where we could mix fully and weakly annotated patients.

[2] trains a network with only bounding box annotations, augmented with DenseCRF. The gist of the method is as follow:
- The bounding boxes are refined directly with a DenseCRF, and used to train the network. As DenseCRF cannot work with binary segmentations, this requires to add an extra hyperparameter: the probability to give to labels (0.7 in place of 1 for instance)
- Those CRF proposals are also used to compute the filling rate of each box (i.e. size of the proposal / size of the box) averaged over the whole dataset.
- The network learns how to "ignore" the outside of the boxes (by modifying the last layer of the segmentation network, and adding a class-wise attention layer that is used to mask the logits output). This, we argue, is akin to a partial cross-entropy, but more complex and potentially less stable.
- Trying to match the average filling rate of the dataset and the predicted filling rate. What they do is to keep only the n% most confident pixels, and supervise them (with cross-entropy). The others, less confident pixels, are ignored.

To work well, the filling rate matching has two big requirements:
- Very good DenseCRF hyperparameters, which is difficult to get for dataset with varying contrast levels (PROMISE12 is problematic in that respecct)
- Some consistency in the object that we want to detect. While this is a reasonable for some organ segmentation applications, this is not for lesion datasets, such as ATLAS, where there is a huge variety of shapes and sizes.

As PROMISE12 is the "simpler" task of the two, we tried [2] on it. We tested two settings: with and without the DenseCRF refining of the bounding boxes. For a fair comparison, we used the same denseCRF hyperparameters as for DeepCut.

The results where not competitive in our application, reaching a maximum DSC of 0.58 without the refining, and 0.65 with it. On top of that, the method proved very unstable, more so when using denseCRF. We found an explanation similar to what causes Deepcut to collapse. On some slices, denseCRF does a good job at refining the bounding box, aligning with image contrast. But, for difficult slices (with very little contrast), the object tended to disappear altogether. This cause unstability, as we end-up training the network with conflicting information. Due to those unsatisfying results on PROMISE12, and the limited time that we had, we did not test it on ATLAS.

We are considering adding this discussion and new results in the final version of the paper. It still requires more work to be included (mostly, replicating the results on PROMISE, and doing the same on ATLAS).

---

### Meta-Review · Area_Chair1 · 2020-04-06
**MetaReview of Paper1 by AreaChair1**

**Rating:** 4
**Recommendation For Accepted Papers:** Oral

**Metareview:**

3 out of 4 reviewers recommended strong acceptance of this work, while 1 reviewer recommended weak rejection. After reading their comments and discussion with the authors, I see that most of the issues raised by the reviewers have been addressed in the authors' response. I therefore think this work can be accepted for publication at MIDL.

Please, when submitting the Camera Ready version, take into account the suggestions made by the reviewers.

**Paper Type:**

methodological development

**Special Issue:**

no

---

### Decision · Program_Chairs · 2020-04-11

Accept